- 1 Alleviating interpretational ambiguity in Hydrogeology through clustering-
- 2 based analysis of transient electromagnetic and surface nuclear magnetic
- з resonance data
- 4 Mathias Vang<sup>1</sup>, Jakob J. Larsen<sup>2</sup>, Anders Vest Christiansen<sup>1</sup>, Denys Grombacher<sup>1</sup>
- <sup>1</sup>Department of Geoscience, Aarhus University
- <sup>2</sup>Department of Electrical and Computer Engineering, Aarhus University
- 7 Correspondence: Mathias Vang (mva@geo.au.dk)

### 8 Abstract

9 10

11 12

17

19

30

Local characterization of groundwater systems is critical for managing and protecting vulnerable resources. Geophysical methods can provide dense imaging of subsurface parameters to delineate lithological boundaries and water tables for hydrogeological investigation. Though, using a single geophysical method for determining lithologies can yield erroneous interpretations as different lithologies can have similar properties. By using several geophysical methods, it is possible to reduce this risk and better assign likely lithologies to subsurface units. We present two case studies where transient electromagnetic (TEM) and surface nuclear magnetic resonance (SNMR) are used in combination to delineate hydrogeological structures. Novel spatially constrained inversion in SNMR was used to provide horizontal consistency between soundings. Three coincident parameters, resistivity from the TEM measurements and water content and relaxation time from the SNMR measurements were used in a K-means clustering scheme to resolve subsurface structures. The K-means clustering was evaluated with a silhouette index to pick the number of clusters. After clustering, each cluster was assigned a hydrogeological description based on the distinct features in the three parameters, e.g. a low resistivity, high water content, and high T<sub>2</sub>\* is assigned as saltwater saturated sand. In the first case study, the clusters enabled improved resolution of a regional water table in an unconfined aguifer setting by the multi-geophysical approach. The water table estimates were positively evaluated against multiple boreholes within 500 m of coincident geophysical models. The second case study illustrates how clustering, of SNMR and TEM models, can delineate saltwater intrusion in an island coastal aquifer, which would not be possible with any of these methods individually. Additionally, the clustering resolved the main shallow aquifer on the island. Our work illustrates how the combination of geophysical data can be used to improve resolution of hydrogeological layers and reduce interpretational bias.

# 1 1 Introduction

2

3 4

11

14

20

2122

23

24

25

26

2728

34

36 Climate-resilient groundwater management hinges on the need for detailed characterization of local groundwater systems (Dragoni & Sukhija, 2008). Historically, lithological descriptions of wells have been used to establish geological models to forecast local groundwater behavior and inform conceptual models of local systems (van Roosmalen et al., 2007). The high cost associated with drilling yields geological maps that are generally based on sparse point coverage, with long-distance interpolation, and simplicity assumptions between observations where structures may actually be complex. To address these data sparsity issues, geophysics can be used to delineate structures non-invasively, giving high resolution imaging of the subsurface to complement direct borehole observations (Binley et al., 2015). Methods based on imaging of subsurface electrical properties are used extensively in hydrological investigations, where spatial variations in the electrical properties of the subsurface, specifically the resistivity, are used to study pollution, explore groundwater resources, and delineate saltwater interfaces, among many other applications (Binley et al., 2015). Within methods imaging electrical properties, electromagnetic (EM) methods are widely used. They operate inductively by creating a varying magnetic field inducing eddy-currents in the ground (Nabighian & Macnae, 1991). The secondary magnetic field produced by the decaying eddycurrents is measured inductively at the surface. The measurements are rapid, which leads to high data acquisition rates that enable mapping of large areas using towed or airborne platforms (e.g. Auken et al., 2019; Sørensen and Auken, 2004). The EM data are translated into 1D models of resistivities by inversion (Christiansen et al., 2006), providing valuable insights into local (hydro-)geology. A limitation of these methods is that they rely on ambiguous links between lithology and resistivity. An implication of this is that local knowledge is required to link resistivity with the associated lithology or geological unit (Dickinson et al., 2010). A common challenge is that different geological units have overlapping resistivity ranges making unique identification based on resistivity alone difficult or sometimes impossible.

Surface nuclear magnetic resonance (SNMR) provides direct sensitivity to water residing in large pores (Hertrich et al., 2007; Legchenko et al., 2002). By transmitting an excitation pulse oscillating at a specific frequency proportional to the Earth's magnetic field strength, the magnetic moment of hydrogen nuclei is shifted from its equilibrium state (Yaramanci et al., 1999). After terminating the pulse, the buildup magnetization decays and is related to the subsurface water quantity and pore parameters. This allows SNMR to track changes in water content across lithological boundaries and can provide valuable information on pore sizes. A limitation in SNMR is the inability to distinguish unsaturated sand from clay, as both will be seen with low WC, in the clays caused by the magnetization decaying extremely rapidly in small pores which makes the clay-bound water undetectable with the SNMR. As such, SNMR

has difficulties distinguishing unconfined aquifers from semi-confined or confined aquifers without supplemental data, as the increase in water content cannot be determined to be a saturation or a lithological transition (Behroozmand et al., 2015), Fig. 1. However, the combined interpretation of SNMR and TEM data, sensitive to different properties, may alleviate ambiguities in distinguishing between for instance unsaturated sand and clays (SNMR ambiguity) or clays and saltwater saturated sands (TEM ambiguity), which is highly relevant for coastal studies of unconsolidated settings (Costabel et al., 2017). Similarly, electrical resistivity soundings and SNMR has been used to alleviate ambiguities in hydrogeological investigations through a joint inversion approach (Günther and Müller-Petke, 2012).

 Consider the example of an unconfined/confined system, where SNMR cannot determine whether a transition from low to high water content marks the water table or a lithological shift from clays to sands. TEM can address this as it would resolve the conductive clay layer if present and delineate the lithological change to sand as seen in Figure 1. If it was an unconfined system, the TEM would image high resistivities in both layers while the saturation change is tracked by SNMR. Another example involves saline intrusion, where TEM cannot differentiate between saltwater saturated sand and clay. If it is indeed a transition only in salinity, not water content, SNMR would reveal continuous high water content across the salinity boundary. SNMR alone would not be able to distinguish freshwater sand from saltwater saturated sand, as it is only sensitive to the abundance of water and not salinity.

Figure 1 Different hydrogeological units resolved with TEM and SNMR. In dashed boxes, only one method is used, and the overlapping units show the ambiguities found.  $T_2^*$  can be implemented to further separate units. Colors in text are not related to colorbars.

1

2

7

12

21

A multiple data type approach requires forming interpretations consistent with multiple geophysical model types simultaneously, which can be achieved through manual inspection of disparate data types. This enables one to distinguish hydrogeological layers through combined interpretation of all data types, but requires subjective choices regarding boundary delineation. Others have used a joint-inversion approach where layer boundaries are set using multiple geophysical methods (Günther and Müller-Petke, 2012; Behroozmand et al., 2012). The joint approaches have the ability to delineate layer boundaries, not seen when inverted separately. An alternative approach employs statistical correlations across separate parameters to partition these into different clusters. One such approach is K-means clustering, which enables the subdivision of datasets based on multiple parameters (Kodinariya & Makwana, 2013). Different clustering approaches have also previously been applied to geophysical data and focus primarily on single source datasets, such as large EM datasets (Dumont et al., 2018) or large electrical resistivity datasets (Song et al., 2010). Some studies investigate clustering on derived parameters such as clay fraction and resistivity, both linked to EM surveys (Foged et al., 2014). Clustering across disparate data types, such as Bouguer anomaly data and magnetic data has been shown to improve the resolution of mineral deposits (Sun and Li, 2016). A study focused on delineating structures in urban settings by clustering on multichannel analysis of surface waves (MASW) and electrical methods to evaluate soil

- foundation structure (Le et al., 2022) and found the K-means clustering to resolve important
- structures in the shallow subsurface.
- In this study, we demonstrate the benefits of combined SNMR and TEM data collection, where
- K-means clustering based on coincident models in two survey areas is shown to enhance
- interpretations and address ambiguities that persist if only a single data type is considered.
- The first example includes mapping of the water table in an unconfined meltwater plain aquifer,
- where a combined approach is used to address ambiguity as to the upper aquifer being
- confined/unconfined/or semi-confined across the investigated region. A second example taken
- from a small island shows how the method can delineate salt-water intrusion from clay-rich
- regions through a combined interpretation. We demonstrate a workflow for handling
- interpretations of SNMR and TEM simultaneously reducing possible interpretational bias.

# 2 Methods

12

13

22

## 2.1 Transient electromagnetic

- 14 In this study we use Transient Electromagnetics (TEM) to resolve subsurface resistivities. The
- 15 tTEM instrument (Auken et al., 2019) was used in both field areas and can resolve the
- resistivity structure of the top 70m, however, here only the top 25m of the full model domain
- are used in the analyses. The induced voltages recorded by the tTEM are translated to 1D
- resistivity models by Spatially Constrained Inversion (SCI) using Aarhusinv (Auken et al.,
- 2015; Viezzoli et al., 2009). The model is discretized into 30 layers with thicknesses varying
- from 1m shallowly, to 10m at depth following resolution limitations at depth. The resulting
- resistivity models will be used for subsequent clustering.

#### 2.2 Surface nuclear magnetic resonance

- In this survey we use a recently developed technique for SNMR called steady-state. The
- steady-state has an increased stacking rate leading to a higher signal-to-noise ratio and a
- decrease in acquisition times (Grombacher et al., 2021). A set of transmit pulses, optimized to
- resolve the top 25 m, was employed in both studies with the Apsu instrument with an
- acquisition time of 25 min per site (Larsen et al., 2020). The resolved water content and the
- relaxation parameter, T<sub>2</sub>\*, are used in the subsequent clustering. The SNMR models are
- discretized into 31 layers down to 50 m, increasing in thickness at depth from 0.5 m to 4 m.
- The resistivity structure from the nearest TEM sounding is used for the inversion. Resistivity
- is needed to obtain the excitation fields used for kernel calculations (Braun and Yaramanci,
- 2008).
- One limitation in SNMR is to detect water residing in very small pores. Because of instrument
- dead times associated with transmitting the excitation pulse (on the order of 8 ms), receiving

- data immediately after pulse termination is not possible. Signals from very small pores can
- therefore partially or fully decay, i.e. lose their amplitude and coherency, before the instrument
- has begun recording data. As such, the magnetization from water residing in very small pores
- decay prior to data recording, which prevents observation of small pore water in SNMR. As
- such, SNMR water contents can be interpreted as a measure of "free" water or an effective
- porosity. T<sub>2</sub>\* relaxation time is linked to pore sizes with low values occurring in small pores,
- while large pores have large values. This can be used to differentiate high water content units
- by their pore sizes.

#### 2.3 Inversion considerations

- Traditionally, 1D SNMR inversions are most commonly treated separately as limited
- measurements are carried out. However, recent acquisition speed-ups enabled by steady-
- state approaches have significantly enhanced spatial data density, which enables the use of
- horizontal constraints linking inversions of nearby measurement sites (Grombacher et al.,
- 2021). One such example is the use of laterally constrained inversion (LCI) for SNMR as
- proposed by Behroozmand et al. (2012) where neighboring sites in a transect can be
- connected. Here, we add a dimension to the constraints using a spatially constrained inversion
- (SCI) framework, not only to bind models in line, but all neighboring models. Delauney
- triangulation is used to find the relevant neighbors as in Viezzoli et al. (2009). The strength of
- lateral bounds is scaled by the distance between models, with a maximum strength defined
- when models are closer than a threshold distance. This threshold distance is typically set to
- the nominal or average distance between neighboring soundings (Vang et al., 2024).
- The computational load increases immensely when implementing SCI with many layers and
- parameters. To reduce the number of iterations, the SCI starting models are defined by single
- site inversion results. This allows the SCI to converge within a few iterations. The TEM data
- are inverted separately with an SCI for the entire survey.

#### 2.4 Clustering

- Large datasets enable statistical approaches to inform on significant hydrogeological units. In
- the following examples, datasets are composed of 50 and 51 coincident SNMR/TEM
- soundings where a K-means clustering is employed (Kanungo et al., 2002) on their model
- parameters. The first step in this type of clustering is to select the number of clusters, K, into
- which the data sets will be clustered (Kodinariya & Makwana, 2013). After selecting the
- number of clusters, the algorithm makes an initial guess for the position of each cluster center
- in the parameter domain. The Euclidean distance from each data point to the cluster center is
- calculated, and each data point is assigned to the nearest cluster. The total distance from all
- data to their assigned clusters is then iteratively minimized through updating cluster center

- locations until either the centroid difference between iterations varies below a set tolerance or
- a maximum number of iterations is reached.
- To improve clustering of datasets for parameters exhibiting different sensitivities and spanning
- different ranges, normalization was used to ensure that each parameter has the same weight
- in the clustering algorithm. Here, we use a Z-score for normalization, where x is either
- resistivity ( $\rho$ ), water content (WC), or relaxation parameter ( $T_2^*$ ):

$$x_{i,norm} = \frac{x_i - \frac{1}{n} \sum_{i=1}^n x_i}{\sigma}$$
 (1)

- where  $\sigma$  is the standard deviation on the cloud of parameters from the inversion,  $x_i$  is the
- parameter value for the  $i^{th}$  data point and n number of data points. Following the normalization,
- we use the Scikit learn package in Python for the clustering and silhouette analysis
- (Pedregosa et al, 2011).
- In this study, the number of clusters is chosen based on the Silhouette index, which calculates
- the membership  $S_i$  of each data point, i:

$$S_i = \frac{b_i - a_i}{\max(a_i, b_i)}, S_i \in [-1, 1]$$
 (2)

- where  $a_i$  is average distance from data point *i* to other data points in the same assigned cluster,
- 16  $b_i$  is the minimum average distance of the i<sup>th</sup> data point to all other data points in other clusters.
- 17 The resulting index, or membership score, is a measure of how well a data point is associated
- with the assigned cluster. If the score of a given data point is 1 it infers that the data point is
- correctly assigned, while a score of -1 indicates that the data is wrongly assigned (Kodinariya
- & Makwana, 2013; Shutaywi & Kachouie, 2021). By evaluating these results, we can qualify
- the preferred number of clusters. The preferred number of clusters is chosen based on two
- criteria. Firstly, the highest average silhouette index indicates that datapoints in general have
- the highest membership score with the given number of clusters. Secondly, we look at each
- cluster and their silhouette index. If more than 50% of the cluster is above the average
- silhouette index, the cluster is well-defined, between 30-50% the cluster is fairly-defined, and
- below 30% it is poorly defined. In some cases, prior information can be used to fix the number
- of clusters, such as prior geological knowledge of the area (Dumont et al., 2018).
- In this study we clustered on three parameters: WC,  $T_2^*$ , and  $\rho$ . The two geophysical methods
- 29 used in this study have different sensitive volumes. SNMR inversion is discretized finely with
- 30 30 layers down to 50 m and the TEM has 30 layers in 120 m. To cluster on coincident values,
- 31 a projection and averaging of the TEM ρ models onto the SNMR discretization is used. All
- TEM soundings within 60 m of an SNMR sounding are included. If there is no  $\rho$  model (TEM

- sounding) within 60 m, the nearest is used and mapped onto the SNMR discretization. This
- 2 allows all SNMR points to be matched and prevents a reduction in data points.

# 3 2.5 Field site description

- 4 Two field surveys were conducted in different geologies to evaluate the use of clustering as a
- 5 tool for alleviating interpretational ambiguity. Both sites were examined thoroughly with SNMR
- and TEM to provide the basis for the subsequent clustering analysis.

### 7 **2.5.1 Kompedal**

- 8 The first field site is Kompedal, a national forest in the Central Region, Denmark. The local
- 9 geology consists of meltwater sand and glacial tills with varying clay contents. The sparse
- borehole coverage finds sand shallowly, and the water table varies from 5 m to 12 m in depth.
- 11 Two geophysical surveys have been conducted here using TEM and SNMR, respectively. The
- scope of the surveys was to delineate the water table on a regional scale and assess whether
- the shallow aquifer can be considered unconfined or semi-confined across the region. The
- 14 TEM data were collected with the tTEM instrument (Auken et al., 2019) while driving along the
- gravel roads within the forest, as seen in Figure 2a in blue. p in the area are generally high,
- above 200  $\Omega$ m, with some layers of lower  $\rho$  found at depth. There is little to no contrast
- between the unsaturated and saturated part of the meltwater sand in ρ. The SNMR survey
- 18 consists of 50 soundings acquired over five days in June 2021, spread across the forest as
- seen in Figure 2a (Vang et al., 2023). The SNMR survey found low WC ( $\sim 5$  %) and low  $T_2^*$
- values (~ 0.1 s) shallowly, with a sharp increase to higher WC (~ 25 %) at 6 m to 10 m depth.
- Layers with low WC and  $T_2^*$  can be associated with both unsaturated sands, and clay-rich
- 22 material. The section indicated in Figure 2a will be used to show the results of the combined
- 23 cluster analysis.

24

### 2.5.2 Endelave

- 25 The second location is a small 13 km<sup>2</sup> island, Endelave, in Kattegat, Denmark with a maximum
- elevation of 8 m. The island's geology consists of glacial till, meltwater sands, and post-glacial
- sands, while boreholes intercept Paleogene clay at depth throughout the island. Generally,
- the glacial tills are found in the west part of the island, where the post-glacial sands are found
- to the north. TEM and SNMR surveys shown in Figure 2b were conducted at this more
- geologically heterogeneous location to resolve possible saltwater intrusion and delineate the
- shallow aguifer found in the meltwater sands and tills. The TEM data were acquired in April
- 2022 and cover the majority of the island and show  $\rho$  below 150  $\Omega$ m for the entire area
- (McLachlan et al., 2025). The ρ resolve buried valley structures and a very conductive
- basement. By TEM alone it is not possible to distinguish Paleogene clay from the saltwater

- 1 saturated sand. The SNMR survey consists of 51 soundings over eight days in July and
- 2 October 2023 and finds high WC shallowly in the east and north part of the island, where the
- 3 west part shows low WC and  $T_2^*$ .

Figure 2 a) The Kompedal survey area. SNMR (red) together with TEM (blue) was collected in the area. b) Map of Endelave survey with SNMR (red) and TEM (blue). Map data: © Google Maps 2024, UTM Zone: 32N.

# 3 Results

# 3.1 Kompedal case study

#### 3.1.1 Clustering analysis

For our combined analysis, we begin by selecting the number of clusters, K, using the silhouette index. Figure 3 shows results from four different clustering analyses with two to five clusters for the Kompedal data set. Each cluster is labeled with its index and the number of data points within each cluster. In each cluster, the silhouette indices are sorted to give a higher index when moving up the y-axis. We use the distinction of well-defined, fairly-defined and poorly-defined, subdivided as mentioned in the results section. In the two-cluster analysis in Figure 3a, we see that both clusters are well-defined with more than 300 members in each and could be a well-suited number of clusters. With three clusters (Figure 3b), cluster 1 is fairly-defined, while cluster 2 is poorly defined with many data points having a below-average silhouette index, and cluster 3 is well-defined. In the four-cluster analysis, two clusters, cluster 1 and cluster 4, become poorly defined, as seen in Figure 3c. Lastly, five clusters yield three poorly-defined clusters (2, 3 and 4) with only few data points having a high membership score. The total average silhouette scores indicated by the grey line highlight that either two or three clusters should be used. Prior hydrogeological information can be used to further qualify the choice between these (Dumont et al., 2018) and in Kompedal, we expect three distinct

hydrogeological units, unsaturated sand, saturated sand and underlying till. The low silhouette indices in cluster 2 in Figure 3b, is a product of large variation within the cluster, which can be expected in glacial environments as mixing occurred during deposition. Finally, three clusters were chosen to subdivide the data into meaningful and decently determined clusters.

Figure 3 Silhouette index analysis for the Kompedal dataset. Four clustering routines were run with different number of clusters, K, (a) two, (b) three, (c) four and (d) five. The sorted silhouette values are shown for each cluster with the average value indicated by the grey dashed line.

Given three clusters, K-means clustering is used to partition the model parameters, WC,  $T_2^*$ , and  $\rho$ . In Figure 4a, the three model parameters are shown in a scatter plot where the color of a point reflects the assigned cluster. The other three 2D scatter plots, Figure 4b, c, and d, show the clustering results projected onto a plane that reveals correlations between two of the three. Cluster 1 in blue, is characterized by a high WC and high  $T_2^*$  value, and a high  $\rho$ . Table 1 shows that large variation occurs within this unit in the SNMR parameters as seen in Figure 4b. The unit is interpreted as a sandy aquifer given its high WC, high  $T_2^*$ , and high  $\rho$ . The very high  $\rho$  (above 300  $\Omega$ m) is a product of very coarse material and that the TEM method can have limited sensitivity to determine resistivity above 150  $\Omega$ m (Christiansen et al., 2006). The yellow cluster 2 has the largest variation in  $\rho$ , hence the low average silhouette index, but generally with lower  $\rho$  values than the other two clusters seen in Figure 4c, and with a large

range in WC. A layer with these signatures is consistent with a saturated sandy till to a more clay rich till, with low  $\rho$ . The overlap with cluster 1 in WC and  $T_2^*$  in Figure 4b is interpreted as a gradual mixing of till and sands. Cluster 3 in red, has low  $T_2^*$  and a low WC and high  $\rho$ , which corresponds to unsaturated sand. However, low SNMR parameters in high  $\rho$  could indicate a silty deposit with smaller pore sizes, but with a similar conductivity. In places where the red cluster is found shallowly, it is interpreted as a unsaturated sand, and at depth under the water table, it is interpreted as a saturated silt.

Figure 4 Clustering results from the Kompedal survey on three parameters: WC,  $T_2^*$  and  $\rho$ . (a) all three parameters in a scatter, (b) WC and  $T_2^*$ , (c)  $\rho$  and WC, and (d)  $\rho$  and  $T_2^*$ . The color of each datapoint defines the assigned cluster; 1-blue, 2-yellow, and 3-red, and their interpreted geology seen in table 1.

Table 1: Cluster parameter bounds and interpreted geology for Kompedal.

| Cluster    | WC          | T <sub>2</sub> * [s] | ρ [Ωm]     | Interpreted  | Label                         |
|------------|-------------|----------------------|------------|--------------|-------------------------------|
|            | [m³/m³]     |                      |            | geology      |                               |
| 1 (blue)   | High        | High                 | High       | Saturated    | SA (Sand Aquifer)             |
|            | [0.1-0.4]   | [0.1-0.4]            | [130-1000] | sand aquifer |                               |
| 2 (yellow) | Medium      | Medium               | Low        | Saturated    | Ti (Till)                     |
|            | [0.07-0.26] | [0.03-0.26]          | [20-300]   | till         |                               |
| 3 (red)    | Low         | Low                  | High       | Unsaturated  | <b>US</b> (Unsaturated sand)  |
|            | [0.04-0.18] | [0.03-0.14]          | [130-900]  | sand         | or <b>Si</b> (Saturated silt) |

### 3.1.2 Spatial interpretation

The three clusters are described in Table 1 and will be referred to by their labels, which are used in the figures to highlight their spatial extent. After assigning interpreted geologies to each cluster, we focus on their spatial position illustrated by a cross-section (location shown in Figure 2a). Consider section 1 in Figure 5, where the coincident data used in the clustering are shown as bars with colors associated with the assigned cluster. The US/Si cluster is situated mostly in the shallow subsurface extending from the surface down to depths of 5 m to 10 m. The grey lines track selected cluster boundaries at the sounding locations. The upper grey line in Figure 5 tracks the bottom of the US cluster and is interpreted to be a change from low-to-high saturation, since the US-cluster is defined by low WC and the underlying clusters have a higher WC. The SA-cluster is found in most soundings and has a variable thickness from 2 m to 17 m. The transition at sounding location 8, is from US to Ti-cluster likely due to lower  $\rho$  in this area. A second deeper grey line tracks the transition below the SA-cluster to the underlying Ti cluster.

Figure 5 Clustering section from Kompedal where the partitioning of data is shown at every sounding. The grey lines track selected cluster boundaries. See Table 1 for cluster descriptions. A is increasing in the South direction.

To evaluate possible variations within the boundaries estimated from the clustering, the profile shown in Figure 5 is reproduced in Figure 6 with  $\rho$  values and WC and  $T_2^*$ . Since clustering is

a discrete and often brutal partitioning of smoothly varying parameters, it is important to return to the original parameters for evaluation. The SNMR WC are shown as bars in Figure 6a and both T<sub>2</sub>\*(left part of bar) and T<sub>2</sub>(right part of bar) are shown side by side with the same color scale in Figure 6b and will be referred to as T<sub>2</sub>\*/ T<sub>2</sub> profiles. The grey lines from the clustering are superimposed on this section to track variations within each cluster unit. Figure 6a displays the first section where shallow low WC and high p coinciding with the US-cluster where the T<sub>2</sub>\*/ T<sub>2</sub> profiles show low values. Boreholes identify this unit as sand near sounding location 3 and 6, which match the interpreted geology as an unsaturated sand. The upper grey line is tracking an increase in WC from ~ 15 % to ~30 % in Figure 6a and from ~ below 0.1 s to above 0.15 s in  $T_2^*$  in Figure 6b, while there is no contrast in  $\rho$ . The lack of structure in the  $\rho$  indicates that the TEM is not sensitive enough to track this saturation change, whereas a lithological change would generally be expected to coincide with a larger p contrast, visible in the TEM data. The elevated T2\* is caused by less interaction with the grain surfaces because of increased saturation in the sand (Falzone and Keating, 2016). Additionally, a borehole water table measurement coincides with this transition line at sounding location 6. The SA-cluster unit contains a range in WC from 20 % to 30 % and in T<sub>2</sub>\* from 0.15 s to 0.3 s, indicating slight variations within the cluster. The SA-Ti transition coincides with a decrease in WC and ρ, interpreted as a similar reduction in pore size, a product of an increase in fines content. The

7

17

- $T_2^*$  found in the Ti-cluster, while quite varying, are generally lower than in the SA-cluster,
- 2 consistent with the interpretation of increasing fines content at depth, as in a till.

Figure 6 Profile of 8 SNMR soundings (bars) and TEM profile(background). Section 1 in Figure 2a. (a) SNMR WC, (b) a split bar with  $T_2^*$  (left) and  $T_2$  (right). Boreholes at sounding location 3 and 6 are shifted ~40 m to avoid overlapping with the bars. Grey lines are tracking transitions between clusters in Figure 5. A is increasing in the South direction.

To further evaluate the accuracy of the ability to track water tables by the upper cluster transition, consider Figure 7, where water tables from clustering are compared to available borehole-measured water tables within 500 m of SNMR sites. The clustering water tables are picked as the transition from the low WC US-cluster to any underlying cluster, SA or Ti, as both have a high WC compared to the US-cluster. The red line has a slope of 1 and the uncertainty bars are equal to the inversion layer thickness at the transition depth, as the clustering method is ternary (i.e., it has three options) and consequently, some layers found at cluster transitions could be assigned to either cluster. We see that clustering tends to overestimate the water table elevation in many cases. This is a product of clustering being a brute thresholding in the parameter space. In this geology, the threshold from the clustering occurs at slightly lower WC than those coinciding with the water table and produces too shallow estimates. The trend, however, is similar to a slope of 1, indicating that a higher threshold could provide a better resolution of the regional water table. Additionally, the distance between borehole and coincident SNMR and TEM models could add uncertainty for the comparison, but this

uncertainty is expected to follow the slope of 1. The two data points with yellow outline, far from the middle axis, stem from the north of the area where the water table was measured in 1980, yielding some uncertainty due to possible long-term temporal or seasonal changes. Overall, the clustering captures the water table trend within an unconfined aquifer at a regional scale in an automated manner.

Figure 7 Borehole water table compared to the clustering water table at 12 SNMR locations with boreholes within 500 m. The red line has a slope of 1 and the error bars on the clustering estimates are based the inversion layer thickness just below the water table estimate to provide a type of uncertainty.

### 3.2 Endelave case study

### 3.2.1 Clustering analysis

As before, we start by selecting the appropriate number of clusters, through silhouette index analyses, shown in Figure 8. Considering we expect a more heterogeneous geology, three to six clusters are used in the analysis. In Figure 8, three clusters are used to partition the data, and result in one well-defined, one fairly-defined and one poorly-defined cluster, whereas the yellow has low and even negative silhouette values, indicating wrongly assigned data points. The average silhouette index is the highest found with the assigned clusters. By using four clusters in Figure 8b, two are well-defined, one fairly and one poorly clustered. We see less negative silhouette index data here, while still maintaining a high average silhouette index. Further increasing the number of clusters to five reveals similar silhouette indexes but has two fairly-defined clusters, however the average silhouette index drops, see Figure 8c. Using six clusters is similar with a few well-defined and fairly-defined, and with a lower average silhouette index. The silhouette analyses show that the number of clusters should either be

three or four as they have well-partitioned clusters, with the highest silhouette index. Prior information from the area indicates that we have four distinct geological units: tills, sand aquifers, Paleogene clay, and possible saline intrusion into sand. The blue cluster in Figure 8b was found to have important hydrogeological information, regardless of its low silhouette index and, as such, we used four clusters for further results.

Figure 8 Silhouette index analysis for the Endelave dataset. Four clustering routines were run with different number of clusters, K, (a) three, (b) four, (c) five and (d) six. The sorted silhouette values are shown for each cluster with the average value indicated by the grey dashed line.

Next, the partitioning of WC,  $T_2^*$  and  $\rho$  is inspected in Figure 9. First, the red cluster (1) is defined by quite low WC and  $T_2^*$  values (Figure 9b), while the  $\rho$  varies from 10  $\Omega$ m to 120  $\Omega$ m. This cluster exhibits properties consistent with till containing varying sand content and affecting  $\rho$  (Figure 9c). The green cluster (2) has mainly high  $\rho$ , high WC, and medium  $T_2^*$  values in Figure 9a. The high WC and  $\rho$  are properties associated with saturated sand aquifers. The yellow cluster (3) has similar SNMR attributes to the red cluster, with low WC and  $T_2^*$ , but has a lower  $\rho$  range illustrated in Figure 9d. This unit is interpreted to be of Paleogene clay due to the very low  $\rho$  found in this cluster. The range of WC found within the yellow cluster indicates that layers with low to medium sand contents, but with low  $\rho$  are grouped here. The last cluster, blue (4) has a distinct  $T_2^*$  range in Figure 9b and a large range

of WC with  $\rho$  situated around 10  $\Omega$ m. The WC and  $T_2^*$  values indicate that this layer has aquifer properties usually associated with sand, while the  $\rho$  indicates this as a conductive material. This is interpreted as saltwater saturated sand. In general, the clusters are not as distinct within the Endelave dataset, as the glacial interaction with the deposited sediment has caused a mixing of lithologies. This is evident from the  $\rho$  values where none exceed 130  $\Omega$ m, whereas the Kompedal survey consisted of  $\rho$  from 50  $\Omega$ m to 1000  $\Omega$ m. All the descriptions and interpreted geologies are found in Table 2.

Figure 9 Clustering results from the Endelave survey on WC,  $T_2^*$  and  $\rho$ . (a) all three in a scatter, (b) WC and  $T_2^*$ , (c)  $\rho$  and WC, and (d)  $\rho$  and  $T_2^*$ . The color of each datapoint defines the assigned cluster, and their interpreted geology, abbreviations seen in table 2.

#### Table 2: Cluster parameter bounds and interpreted geology for Kompedal.

| Cluster   | WC [m <sup>3</sup> /m <sup>3</sup> ] | T <sub>2</sub> * [s] | ρ [Ωm]   | Interpreted geology | Label           |
|-----------|--------------------------------------|----------------------|----------|---------------------|-----------------|
| 1 (red)   | Low-medium                           | High [0.02-          | High     | Till                | Ti (Till)       |
|           | [0.03-0.18]                          | 0.1]                 | [10-120] |                     |                 |
| 2 (green) | High                                 | Medium               | High     | Sandy aquifer       | SA              |
|           | [0.15-0.40]                          | [0.04-0.13]          | [15-120] |                     | (Sand aquifer)  |
| 3         | Low-medium                           | Low                  | Low      | Paleogene clay      | CI (Clay)       |
| (yellow)  | [0.03-0.18]                          | [0.02-0.1]           | [1-25]   |                     |                 |
| 4 (blue)  | Low-High                             | High [0.07-          | Low      | Saltwater sands     | Sws (Saltwater  |
|           | [0.07-0.40]                          | 0.21]                | [2-35]   |                     | saturated sand) |

### 3.2.2 Spatial interpretation

Following the clustering we will examine their spatial extent on Endelave. We will show the results of two sections (Figure 2b). to see how the clustering performs in a more heterogenous setting. Consider first the section across the main shallow aquifer in Figure 10a, where we see a shallow Ti-unit corresponding to either a till or unsaturated sand. The SA-cluster unit has a thickness from 5 m to 12 m and is found below the Ti-cluster at sounding locations 3 to 6. Sounding location 1 is located 30 m from the coast, which aligns with the presence of the Swscluster. The Ti-cluster at depth is interpreted as a decrease in pore size from an increased clay or silt content. At sounding 7, all layers are grouped as the Ti-cluster, a sign of low SNMR parameters throughout the entire sounding location. The deepest discretized layers at most sounding locations are grouped in the CI-cluster, tracked by a grey line, indicating a drop in  $\rho$ , as expected from the Paleogene clay.

To highlight possible saltwater intrusion, a section intersecting sounding locations at the coast is shown. The section in Figure 10b is quite complicated as it transects different geological regions. We consider three main points in this section, the Sws-cluster, the SA-cluster and the south end of the profile. In Figure 10b we see the Sws-cluster at sounding locations 1 to 3, defining a shallow and deep layer, while at sounding locations 6, 8 and 9 the cluster is seen shallowly at low elevations following the coast. The transition from the Sws-cluster to the underlying clusters is tracked by a grey line at sounding locations 1 to 3. Below the grey line at sounding locations 1 and 2, the layers are grouped with the CI-cluster representing low  $\rho$ , lower  $T_2^*$  and WCs. It is important to note that even with combined SNMR and TEM, it will be

- hard to distinguish between saltwater and freshwater clays as both will be conductive and
- have a low free water content and T<sub>2</sub>\* signatures in SNMR.

- From sounding locations 3 to 8, the SA-cluster is found with a varying thickness from 2m to
- 10m. This unit, interpreted as the aquifer, is outlined in grey to compare with original parameter
- values and borehole information later. At the south end of the profile, the clustering divides
- layers into CI- and Ti-clusters, associated with clay and till by their low WC and T<sub>2</sub>\*.

Figure 10 Clustering sections from Endelave where the partitioning of data is shown at every sounding location. (a) Section 1 (b) Section 2 in Figure 2a. The grey lines track selected cluster boundaries. A is increasing in the Southeast direction. B is increasing in the South direction.

The discrete boundaries from the clusters are now used in the original parameter space to evaluate possible variations within the clusters and the estimated boundaries. In Figure 11, we consider the main shallow aquifer found on Endelave. The grey lines from Figure 10a are used to delineate cluster extents and each unit is assigned a cluster label. Shallowly, the Tiunit coincides with low WC and a high ρ in Figure 11a.  $T_2^*/T_2$  are low in this unit and boreholes reveal either till or unsaturated sand here, matching the clustering interpretation. The upper Ti-SA grey line tracks an increase in WC at four sounding locations and coincides with a lithological change from clay to sand in two boreholes and coincides with a water table measurement in a separate borehole. This is interpreted as a semi-confined system with the water table coinciding with a lithological layer boundary. The SA unit here consists of high WC

and low to medium  $T_2^*$  within a resistive unit. The boreholes identify this unit as sand or a mixture of sand and silt, which explains the range of WC grouped within this unit. The lower SA/TI transition tracks a decrease in WC, still with low to medium  $T_2^*$  seen in Figure 11b. The transition coincides with a decrease in  $\rho$  at sounding locations 1 and 2, and with a lithological boundary from sand to clay in a few boreholes. Furthermore, two boreholes terminate exactly at this interface, which could indicate that the drillers hit something harder or more clay rich, prompting them to stop drilling. The Ti/CI transition at depth tracks a decrease in  $\rho$ , which in the deep borehole is identified as a lithological boundary from clays and sand to Paleogene clay, agreeing with the clustering interpretations. This deep boundary is not seen in the SNMR-parameters as the Ti and CI-clusters are only distinguishable by their  $\rho$ .

Figure 11 Profile of 7 SNMR soundings (bars) and TEM  $\rho$  (background). Section 1 in Figure 2b. (a) SNMR WC (b) a split bar with  $T_2^*$  (left) and  $T_2$  (right). Grey lines are tracking transitions between clusters in Figure 10. A is increasing in the Southeast direction.

After reviewing the section through the main shallow aquifer in Figure 11, we will assess a second, more complex section. The grey lines from Figure will be used to delineate the cluster units and illustrate differences within the units. Consider now Figure 12, where the  $\rho$  and SNMR parameters are shown with lines following cluster transitions. The Sws unit is seen mainly at location 1 to 3 and is defined by high WC, very high  $T_2^*$  and low  $\rho$ . At sounding location 8, a borehole finds sand coinciding with the Sws-cluster in agreement with the

saltwater saturated sand interpretation. The high  $T_2^*/T_2$  associated with the Sws-cluster in Figure 12b is a product of limited compaction within the newly deposited sand in the coastal environments. Below the Sws-unit, the grey line tracks a transition to lower WC and  $T_2^*$ , but maintaining the low  $\rho$ , which is defined by the CI cluster. At sounding location 6, this transition is different with an increase in  $\rho$  tracking the border to the SA unit. Low WC at sounding locations 10 and 11 coincide with clay in a local borehole for the first 15 m where all layers are grouped within the Ti or CI-clusters. The low water content and  $T_2^*$  signature at these locations prevent them from being clustered with the saltwater saturated sands in Sws, highlighting the value of SNMR to distinguish these conductive units. The gyttja layer found in the borehole coincides with a drop in the SNMR WC due to the increases in organic matter, decreasing the pore size, and was grouped with the CI-cluster (Mashhadi et al., 2024).

Figure 12 Profile of 11 SNMR soundings (bars) and TEM  $\rho$  (background). Section 2 in Figure 2b. (a) SNMR WC (b) a split bar with  $T_2^*$  (left) and  $T_2$  (right). Grey lines are tracking transitions between clusters in Figure 10. B is increasing in the South direction.

By clustering on this dataset, we have proven the ability to identify regions of possible saltwater intrusion. Figure 13 shows which sounding locations have layers that cluster within the saltwater aquifer, the freshwater aquifer, that have layers of both clusters, or only have the till and clay clusters. The saltwater cluster is observed mostly at the northern sounding locations where the post-glacial sands are located, but also along the east coast. The main

aquifer unit, SA, is found in the east and north parts of the island, while the west part is dominated by the low water content clusters, shown in yellow and red. One sounding location with both saline and freshwater clusters far from the coast, is observed in the north of the island. The closest TEM sounding was acquired in a lowland south of this sounding, with elevation almost at sea level, which might cause issues with saltwater intrusion. There is also a wetland close to this location, which might have a higher clay content with low  $\rho$ . If the TEM and SNMR are not exactly coincident, some differences and anomalies in the clustering might occur. But in general, the K-means clustering is able to map this possible saltwater intrusion, which is a valuable asset in aquifer management.

Figure 13 Sounding locations where salt water or fresh water has been identified. Locations with only clay and till clusters are shown with red and yellow. Map data: © Google Maps 2024, UTM Zone: 32N.

## 4 Discussion

 In this study we investigated the use of clustering to combine the analysis of two geophysical methods, SNMR and TEM. The K-means clustering was found to be able to differentiate units into interpretable hydrogeological layers and was consistent with manual interpretations. Combining the datasets helped alleviate some of the ambiguities found when interpreting based only on a single dataset, i.e., unsaturated/confined conditions in Kompedal, and saltwater/freshwater in Endelave.

1 K-means clustering on geophysical models offers a simple, automated approach to identifying

lithological transitions. It allows for reproduceable boundary definitions without subjective

interpretations of the geophysical models. Discretizing smoothly varying parameters into

predefined clusters is, however, brutal and there will be variability within the unit definitions.

5 The ability to return to the original parameter space with cluster boundaries is crucial in

addressing subtle variations within units and can be used to evaluate cluster transitions.

K-means clustering applied to geophysical models is not limited to SNMR and TEM parameters; it can also be extended to other collocated datasets with distinct sensitivities. For example, in areas where a deep water table is expected within a sand layer, seismic methods may be appropriate. However, because the seismic velocity of saturated sands can be similar to that of clays or tills, incorporating collocated TEM models can help reduce interpretational bias. Similarly, relying solely on TEM data may make it difficult to detect the water table due

to limited sensitivity to high-to-high resistivity contrasts.

In SNMR, correlations between WC and  $T_2^*$  may exist (Falzone and Keating, 2016). For example, in unsaturated sands, the low water content residing in the pores will be in close contact with the grain surfaces, resulting in interactions leading to low  $T_2^*/T_2$ . Since water content is proportional to signal amplitude, in low WC environments, low signal amplitude results in reduced confidence on the  $T_2^*$  estimates. When such parameters are linked, it might be of interest to simplify the approach by clustering on the product of water content and  $T_2^*$ . Thus, combining these two parameters may help reduce the influence of low-confidence  $T_2^*$  values in low water content environments. A similar option is to use a principal component analysis to reduce the basis to two parameters that describe most of the variance, which in the Kompedal case would be resistivity and the product of the SNMR parameters. However, in more complex geologies such as Endelave, a decrease in basis dimension may reduce the ability to distinguish layers of high WC and low  $T_2^*$  from layers with low WC and high  $T_2^*$ . Through examining the data's variation and correlation, we can make qualified decisions about whether to decrease the parameter space.

In this study, we focused on interpreting two survey areas using K-means clustering, which proved sufficient in meaningfully partitioning data and identifying lithological boundaries. One feature of the employed K-means clustering approach is the need to specify clusters beforehand. In this study we based the choice of clusters on the silhouette index and prior geological information about the area. One alternative study uses agglomerative hierarchical clustering on SkyTEM data, which avoids selecting the number of clusters by starting with one cluster and subdividing until each data point has its own cluster (Dumont et al., 2018). This

can alleviate some of the choices made for the silhouette index analyses and provides a better understanding of how clusters are further subdivided. A second challenge is to attribute uncertainties to the layer boundaries picked by the discrete K-means clustering. Here, others use fuzzy C-means where data points are assigned a membership score and can be partial members of more than one cluster (Paasche et al., 2007). Applying the fuzzy C-means can give an estimate of uncertainty for the picked cluster boundaries, i.e., if a data point could be a member of several clusters, it is less certain. This could apply to the Endelave data where the saline intrusion cluster in places could overlap with the freshwater cluster.

Another way of exploiting collocated datasets is the use of joint inversion for layer boundary picking. Studies identifying layers from SNMR and TEM implementing various regularization techniques has shown promise in reducing the ambiguity found when interpreting each separately (Behroozmand et al., 2012); Skibbe et al., 2018). These approaches focus mostly on the collocated datasets and invert these jointly. In our study, the tTEM data is inverted separately with the full survey of more than 23000 datasets. As such, we have the ability to track the changes in resistivity in places where the SNMR is not present. Additionally, the framework for using joint inversion in steady-state SNMR is not established as kernels are calculated before the inversion, fixing the discretization. Further investigations could focus on implementing clustering in a joint inversion framework with a large spatial extent. This could alleviate some of the interpretational load when dealing with large datasets.

Since clustering is performed on coincident values, we are limited by the lowest dimension dataset, which in this case is the SNMR, e.g. on Endelave the survey consists of 51 soundings while there are over 23000 TEM soundings in the same area. This reduction in data space disregards large amounts of TEM data, which of course have valuable regional information, but lack coincident SNMR parameters. Additionally, lower data quantity can lead to clusters not representable for the area. If SNMR information could be extrapolated to the full TEM domain through appropriate spatially variable measures, it would allow for clustering on a much larger data set. Future research will focus on extrapolating SNMR parameters across the full TEM domain. This would enable a subdivision of the full TEM domain based on the coincident data clusters, and it will be possible better to delineate areas of potential saline intrusion spatially.

As there is limited ground truthing information, the clustering has been mostly compared to manual interpretations based on the data collected. This does not directly provide validation of the layers seen but infers that the clustering is performing like expert interpreters would if given a similar data set. As such this study shows the value in having clustering as the main subdivider of lithological units instead of having manual inspection of each collocated dataset.

- 1 Given the recently enabled larger scale mapping with SNMR, a less subjective and fast
- 2 interpretation scheme is a step towards automation from data to lithology.

# 3 5 Conclusion

- Through two field studies we demonstrated the automated spatial identification and separation
- of hydrogeological units in large scale geophysical campaigns. Recent improvements in the
- data acquisition rates of SNMR now offer data volumes sufficient to exploit clustering
- approaches when combining these data with other geophysical data. K-means clustering of
- complementary SNMR and TEM models is shown to provide a less-subjective approach,
- where enhanced hydrogeological interpretations can be formed by exploiting the
- complementary nature of two data types. To detect lithological boundaries, they must
- correspond to a contrast in geophysical properties. SNMR is shown to provide value when
- discriminating clay-rich sediments from saline saturated sand conditions, a challenging task
- based on only TEM models. Similarly, TEM is able to separate low-water content conditions
- 14 from clay-rich conditions, which is impossible with SNMR alone. This is key to discriminating
- 15 between unconfined and semi-confined conditions. A silhouette index-based approach,
- 16 combined with the a priori knowledge of the likely number of lithological units present, was
- used to select the number of clusters and found to be suitable for these datasets.
- In the examples, clustering of NMR and TEM data provides a more complete characterization
- of local hydrogeological conditions than what can be achieved by each data set separately.

## 20 Data availability

21 The data shown in this study are available upon request from the corresponding author.

### 22 **Author contributions**

- 23 MV wrote the manuscript, gathered data and did the analysis. JJL helped with writing the
- 24 manuscript. AVC helped with the analysis and correcting the writing. DG assisted in figure
- development, analysis and writing the manuscript.

### Competing interest

27 The corresponding author declares that none of the authors have any competing interests.

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
