# Peer review of "Alleviating interpretational ambiguity in Hydrogeology through clustering-"

_EGUsphere, 2025_

## Referee Comment (RC2)

Review of

Alleviating interpretational ambiguity in Hydrogeology through clustering-based analysis of transient electromagnetic and surface nuclear magnetic resonance data

The work that is presented in this manuscript is an important step towards an automated and thus fast hydrogeological interpretation of combined TEM and SNMR measurements. This combination has been applied in previous studies and has been proven to be powerful, especially within the frame of groundwater characterization in coastal areas, where the risk of relevant saltwater intrusion appears and frequently generates problems of freshwater production. In this environment, advanced knowledge of the aquifer situation, hydrogeologic modelling and temporal observation of the groundwater quality is desired and can most oftenly not be realized by drilling boreholes alone.

Recent improvements of TEM and SNMR have enabled options to produce big datasets that need to be interpreted in time to gain real social, economic and ecologic relevance in supporting freshwater production. Consequently, automated approaches such as the one presented in this study are the next logical step to deal with such data. The paper represents a significant contribution within the field of hydrogeophysics and its application and consequently fits in very well with the HESS journal.

The manuscript is well structured and the principle path to the results and conclusions is easy to follow. The figures are well presented and helpful to understand the key messages. However, some important technical information and discussions are missing, especially on details of the implementation and performance of the K-means approach. As I understand the study, the focus is not on the verification of the approach, e.g. by concrete ground truthing which is difficult given the large spatial extent covered by the surveys, but rather the introduction of the principal strategy that has still potential to be further developed. Please read my detailed concerns and recommendations below.

In the presented case studies, the outcome of the clustering analysis is basically tested against the classical manual interpretation at the basis of the original data with only a few observation boreholes. These are not sufficiently distributed over the profiles to enable reliable validation to the full extent. Consequently, the discussion section should honestly admit that there are still significant uncertainties, which cannot be resolved for several reasons. Please find details below.

However, the focus of this work and the value of automated clustering approaches in general is not on interpreting local situations as accurately as possible, even if the data at the specific position possibly allows this, but on providing a reliable overview at a larger scale. In this sense, I want to encourage the authors to include such reasoning in their motivation and discussion. Again, please find details below.

Finally, I assess the necessary revisions of the manuscript into the cathegory moderate-to-major.

**Details**

**P2L12-13:**

… electrical properties of the subsurface, specifically the resistivity, are used …

**P2L22:**

A limitation of…

**P2L23:**

The sentence starting with „An implication of …" is overloaded with redundancy and word repetitions. Please reformulate.

**P3L1:**

„…making it immeasurable with…" => „…which makes the clay-bound water not detectable with…".

**P3L3:**

„established" => „determined"

**P3-Fig1:**

I like the idea of this scheme. However, I do not see the necessity of providing the axes as colorbars. It is rather confusing – I tried without any success to relate the font color inside the scheme to the colorbars of the axes. If you insist of using them, please clarify in the figure caption how they should be understood. Otherwise, I suggest avoiding them.

Please change the label of y-axis: „SNMR water content" instead of „water content" to avoid confusion or misconception. As you correctly describe in the main text, clay normally exhibits high water contents, but apparently appears as low or zero water content in SNMR.

**P5L5:**

Incorrect statement: the increased stacking rate leads to higher singal-to-noise ratios, not to higher signal amplitudes.

**P5L11:**

Clarify with an additional brief sentence and a corresponding reference, why resistivity information is necessary for SNMR inversion.

**P6L13:**

I am not familiar with K-means approaches. However, as far as I know, the performance of any optimization algorithm strongly depends on the chosen termination criteria. Please explain in more detail, here or later in the main text, how your approach actually performes. How many iterations were chosen exactly? Did you reach its maximum number for the majority of the runs or did the algorithm converge properly to the predeterimed minimum distance? Can you quantify the minimum distance? Are there any criteria how to choose the minimum distance or is it rather a try-and-error process to find a proper threshold?

**P7L8 (subsection 2.5)**

This subsection, in my opinion, should appear at the very beginning of the methods section. Explain first what you did and where, before going into detail on each measurement method and processing approaches.

**P8L2**

Reading about „TEM soundings" here is rather confusing – it indicates that you conducted the „traditional" way of applying TEM. As I understand it, tTEM is rather a continuous sequence of TEM measurements while driving through the area. Please clarify.

**Subsections 2.5.1 and 2.5.2:**

You already provide results, their interpretation and some conclusions here, while introducing the investigation areas. This is Okay in principle, if all this information has already been puplished. In this case, please specify the corresponding references. Otherwise, these statements belong to Section 3.

**Subsection 3.1.1 (on the description of the silhouette index analysis and interpretation of Fig. 3)**

Unfortunately, some explanations in this passage remain unclear to me.

What exactly is the average silhouette index? Following the main text, it seems to be calculated for each cluster separately. But this is in contrast to the gray dashed lines in Fig.3 that represent a threshold for the whole data points in each subfigure. Moreover, it seems to be the very same value for each subfigure, no matter how many clusters are chosen - Is this plausible? Please clarify.

Furthermore, when having a closer look at Fig.3, I cannot follow the classification of „well-, fairly and poorly defined" that is made in the main text. It states for instance that cluster 1 in Fig.3b was well-defined (P8L22-24) but more than half of the data points in that cluster are below the average value, some points even appear with a silhouette index below 0. Following the definition in the main text this should be classified as a poorly defined cluster (P8L20: „…many data points below the average…"). Similar confusion appears for the interpretation of the other subfigures. However, maybe I had missed something relevant and completely misunderstood this analysis. Please clarify and reformulate the text accordingly.

**P10-Fig4**

Please specify the appreviations of the legend entries in the figure caption or provide a reference to table 1.

**P11-Fig5 (and the other figures with profile data)**

Please consider indicating <N>orth and <S>outh directions at beginning and end of the profile(s) to make it easier to follow the interpretation(s) in the main text.

**P12L11**

Following Archie's law, a change in water saturation should also lead to a contrast in rho. The fact that the change of water content measured by SNMR is not indicated by the TEM data is most likely due to the fact that TEM is not very sensitive to rho contrasts above 100 Ohm*m. This information should be given beforehand together with a suitable reference, e.g. when describing the scheme in Fig.1.

**P12L13**

I do not doubt this statement, but you should provide a reference to verify it. There are some papers around that study the relationship between water saturation and relaxation time for loose sediments.

**P13-Fig6 (same for Fig.10 and 11)**

I recommand focusing on T2* alone in the lower subfigure(s) to avoid confusion. My first reaction was wondering what additional information the ratio of T2*and T2 could provide. I guess there is no additional information in the T2 parameter compared to T2*, right?

**P13L14**

Please describe more in detail how the error bars in Fig.6 must be understood and how the uncertainty in assigning the data to the clusters affects the uncertainty of the water table estimation. What means „the uncertainty bars are based on the layer thickness"? Is it identical with the layer thickness? In this case, one could assume that, in general, we just have to decrease the layer thicknesses in the inversion model to increase the accuracy of the water table estimation. This would be nonsense, of course!

Sorry for stressing this aspect, but the confidence of non-invasive geophysical water table estimates is, in my experience, a frequent question in technical discussions with hydrogeologists. Your Figure 7 could be a key figure for the combined TEM/SNMR approach in this regard. However, it is not yet clear how to interpret it exactly.

And there is another aspect to consider when interpreting Fig.7. The water table is never identical with the interface between saturated and unsaturated zones. The saturated zone is always above the water table due to capillary forces. Depending on the lithology the difference between the two varies between a few cm (coarse sand) to several meters (till). Is such an interpretation of the descrepancy between the data points and the 1:1 line also reliable, given the actual geology?

**P13L21 and Fig.7**

I strongly recomment erasing the two yellow points from the crossplot. The attempt to verify the recent water table estimates with completely outdated information has no scientific value, even if it matched the 1:1 line by chance - which it obviously does not!

**P13L12**

Why do you not show the silhouette index plot for the Endelave site as well? Even if it was more complicated and difficult to interpret than for the case in Fig.3, the reader could learn a lot from it! How objective can the choice of the number of clusters even be? If an objective approach fails, we have to discuss criteria for subjective choices. As the number of clusters is crucial for this kind of analysis, I would expect a more detailed discussion on it for the two cases presented in this paper.

**P13L13**

I recomment to erase the sentence „Consider Figure 8,…". The wording is technical incorrect, because the clustering results cannot be represented by two parameters. It is always three parameters regardless of how they are depicted. However, the sentence is not necessary at all.

**P14L17**

The red cluster might also include unsaturated sand or can you exclude this possibility for some reason?

**P14L19**

saturated sand aquifers

**P14L22**

„could indicate" => „indicates"

**P15L2**

„would indicate" => „indicates"

**P15-Fig8**

Please specify the appreviations in the legend.

**P16L5**

Okay, we learn later that unsaturated sand at this position is somewhat unreliable because of the borehole information. However, I miss a discussion in the paper about the resolution properties of the tTEM at shallow depths < 10 m. This is relevant for the interpretation of the near-surface till layer. TEM does obviously not resolve it, although one would also expect low resistivities here.

**P17L8**

Figure 9a

**P17L15**

What does „due to shifts in geological deposits" mean in this context? Please explain more in detail.

**P17L20**

A few boreholes? There is only one borehole in the vicinity of sounding 1 and 2. Furthermore, I would expect a decrease in rho at the SA-TI interface. Again, this contrast can obviously not be resolved by the tTEM method in this specific case. However, in Fig. 6 we see that it resolves a rho contrast between SA and TI even at a similar depth. Why not here?

**P18-Fig10**

The reddish colors of Silt and Clay/till are difficult to distinguish in the figure. Please use another color for one of them.

**P19L4**

Please be precise! The organic matter does surely not lead to a drop in WC (the opposite is the case!), which is very obvious when inspecting the cited paper of Mashhadi et al. (2024). Similar to clay - also with gyttja it is actually not a „real" but an apparent drop in WC caused by the fact that short $T2^*/T2$ times are not detectable with SNMR.

**P19L18**

„might have" => „might cause"

**P20L8 – P21L6**

This passage is redundant with the descriptions in the introduction and can be removed from the discussion section completely. Please only discuss here what is new and implied by the direct results of your study.

**P21L20**

As already mentioned (comment on **P12L13**), there are references to verify this statements. This is not a conclusion or finding of your study.

**P21L23**

„confidence in" => „confidence of"

**P2L26**

„reduce" => „reducing"

**P2L28**

„describe the most variance" => „describe most of the variance"

**P21L32**

„informed" => „qualified"

**P22L5**

provides

**P22L29**

I strongly doubt that the suggested approach is strictly „non-subjective", because of the crucial predefinition of the number of clusters (please see my comments on the silhouette index approach). Use „less-subjective" or a formulation such as „towards a non-subjective interpretation".

**P23L3**

I disagree with this statement! Only for one of the site, the silhouette index approach was introduced and described, at all, and even this is hardly comprehensible (see my comments on section 3.1.1). For the second site, the choice of the number of clusters seems rather arbitrary to me. Finally, no concluding remarks can be made about the robustness of the suggested approach. This is also true for the whole K-means algorithm as it is used here and remains an objective for future research. The robustness could, for instance, be analysed in a pure synthetic parameter study.

---

## Author Comment (AC1)

Reviewer 1

Thanks for tacklig an important question in hydrogeophysics with your manuscript. I went through with pleasure. It is written straight forward, thus easy to follow. Figures are well presented.

Nevertheless I have some comments basically on the methodology. I hope they help to clarify/sharpen the main purpose of the manuscript, solve some unclear interpretation and maybe extending the research context.

To me the main purpose of the manuscript is the (more objective and formalized) cluster analyses enabled by a larger number of SNMR sounding compared to "just" manual interpretation of multiparameter geophysical data. However, I think, this part needs sharpening and clarifications.

Author response: Thank you for your comments. We have followed your recommendations and believe the manuscript have been significantly improved. Especially the comments on the clustering formulation has been clarified as well as adding some discussion on joint inversion vs clustering.

--- Komedal:

-Figure 3 gives the silhuette index for the clusters and the text introduces the terms "well defined", "fairly defined" and "poorly defined" that are used to decide for the 3-cluster case. First, the definitions of the terms is vague. For instance, "well defined" is defined by "most of the data ...". What does "most" mean? Give better definitions.

Author response: We have added a more thorough description of the silhouette. We have added the average silhouette index to the figure for each of the cluster sequences. We are interested in having both well defined clusters, while maintaining a high average silhouette index, which is the case for the Kompedal 3 cluster case. We have added the average silhouette index to the silhouette figures to better highlight this.

P7.L21-L26:
 *"The preferred number of clusters is chosen based on two criteria. Firstly, the highest average silhouette index as indicates that datapoints in general have the highest membership score with the given number of clusters. Secondly, we look at each cluster and their silhouette index. If more than 50% of the cluster is above the average silhouette index, the cluster is well-defined, between 30-50% the cluster is fairly-defined, and below 30% it is poorly-defined."*

-Further, to me it is unclear why the 3-cluster case is taken. That case contains a "poor defined" cluster! Taking the siluette analyses I would go for two clusters. However, it is argued with lithological knowledge one would take 3 cluster and indeed when looking at Figure 4 it very much looks like 3 cluster. But this is not the outcome of the siluette analyses.

Author response: It is true that when looking only at the silhouette values for single clusters the two-cluster case looks better. We have added more description on how we take the total average silhouette index into account. In the Kompedal case, we have added more description on the clustering results.

*"For our combined analysis, we begin by selecting the number of clusters, K, using the silhouette index. **Error! Reference source not found.** shows results from four different clustering analyses with two to five clusters for the Kompedal data set. Each cluster is labeled with its index and the number of data points within each cluster. In each cluster, the silhouette indices are sorted to give a higher index when moving up the y-axis. We use the distinction of well-defined, fairly-defined and poorly-defined, subdivided as mentioned in the results section. In the two-cluster analysis in **Error! Reference source not found.**a, we see that both clusters are well-defined with more than 300 members in each and could be a well-suited number of clusters. With three clusters (**Error! Reference source not found.**b), cluster 1 is fairly-defined, while cluster 2 is poorly defined with many data points having a below-average silhouette index, and cluster 3 is well-defined. In the four-cluster analysis, two clusters, cluster 1 and cluster 4, become poorly defined, as seen in **Error! Reference source not found.**c. Lastly, five clusters yield three poorly-defined clusters (2, 3 and 4) with only few data points having a high membership score. The total average silhouette scores indicated by the grey line highlight that either two or three clusters should be used."*

-Finally it is agrued that cluster 3 is interpreted differently when being shallow or deep (page 10 lines 7 to 10). To me this "additional" interpretation to the cluster analyses is really weakening the more objective and automatic approach.

Author response: Thank you for the comment. This is indeed less objective. From the parameters presented, it is not possible to distinguish these different lithological units, unless more information is added, such as the water table to distinguish the unsaturated from saturated material. A more appropriate interpretation would be that the red cluster is a low-free water unit with high resistivity. For non-experts this might be too specific, which is why we opted for two interpretations. Another option would be to include the depth in the clustering. But as we tried only to include inverted parameters in this study we left out fixed parameters.

-It is argued that the cluster analyses is somehow a "brutal" approach for a smooth inversion. But then, why not using a layered inversion?

Author response: The smooth inversion is used as we do not have the flexibility to use a layered inversion with the current modelling setup for steady-state sequences used for SNMR. Here, we are bound to a fixed discretization stemming from the kernel calculations to limit computational time.

-In particular I am also suprised by the missing contrast of resistivities between saturated and unsaturated. Water saturation is a driver of the bulk resistivity. If saturation changes, resistivity changes! Is that maybe because of missing sensitivity of "inductive electromagnetic methods" to changes in "higher" resistivities, thus parameter uncertainties for rather resistive layers? It is surely not because of inability in the physical parameter as written for instance at page 12 line 10 or page 21 line 18-19. Electrical methods are used to monitor the vadose zone and water infiltration. Please be more clear. And not at least, TEM is used to monitor groundwater table! (Zamora-Luria et al 2024, Long-term monitoring of water table and saltwater intrusion ..., Near Surface Geophysics, Vol. 22). However, the cluster analyses solves that "problem" nicely into 3 clusters (because of the SNMR sensitvity to water content), thus, this seems to be the advantage of the cluster analyses. Please elaborate on this.

Author response: Yes, indeed the water saturation is a driver of the bulk resistivity. But as the saturated and unsaturated material has resistivities over 150ohmm, it is difficult to separate these in TEM as most sensitivity is below this. Electrical methods could perhaps be a more suited to separating these two in this case. Yes, TEM has been used for groundwater table monitoring but relies on very fixed inversion setups to catch these minimal changes as well as longer stacking as the system is stationary. Especially the discretization has to be matched accordingly, otherwise the water table might be smoothed out lying not at a layer bound but within a layer.

We have rephrased the sentences in the paper to more accurately describe, that it is the lack of sensitivity to changes in high resistivity of TEM, and not the lack of change in resistivity between the saturated and unsaturated media.

P16L5-L7

*"The lack of structure in the ρ indicates that the TEM is not sensitive enough to track this saturation change, whereas a lithological change would generally be expected to coincide with a larger ρ contrast, visible in the TEM data."*

P21L18-19

*"Similarly, relying solely on TEM data may make it difficult to detect the water table due to limited sensitivity to high-to-high resistivity contrasts."*

--- Endelave:

As for Komedal, please show the siluette plot for different numbers of cluster.

Author response: The silhouette plot for Endelave has been added and described.

P19L3-P19L19

"As before, we start by selecting the appropriate number of clusters, through silhouette index analyses, shown in **Error! Reference source not found.**. Considering we expect a more heterogeneous geology, three to six clusters are used in the analysis. In **Error! Reference source not found.**, three clusters are used to partition the data, and result in one well-defined, one fairly-defined and one poorly-defined cluster, whereas the yellow has low and even negative silhouette values, indicating wrongly assigned data points. The average silhouette index is the highest found with the assigned clusters. By using four clusters in **Error! Reference source not found.**b, two are well-defined, one fairly and one poorly clustered. We see less negative silhouette index data here, while still maintaining a high average silhouette index. Further increasing the number of clusters to five reveals similar silhouette indexes but has two fairly-defined clusters, however the average silhouette index drops, see **Error! Reference source not found.**c. Using six clusters is similar with a few well-defined and fairly-defined, and with a lower average silhouette index. The silhouette analyses show that the number of clusters should either be three or four as they have well-partitioned clusters, with the highest silhouette index. Prior information from the area indicates that we have four distinct geological units: tills, sand aquifers, Paleogene clay, and possible saline intrusion into sand. The blue cluster in **Error! Reference source not found.**b was found to have important hydrogeological information, regardless of its low silhouette index and, as such, we used four clusters for further results."

I found the parameters and hydrological interpretation in the clusters somewhat suprising. Figure 8 indicates 4 cluster, no doubt,  but especially the T2* times of sandy aquifer are quite low and the range between "clay", "till" and "sand" similar while for the saltwater saturated sand (saline sand is not a good term) T2* is higher? Why is that? Is that related to "impacts" (iron) on the T2' relaxation times? Explain! However taking a look at the spatial distribution the clusters are reasonable, so why is that? Apparently the cluster analyses works well and help identifying units. So this could be a real benefit. Please elaborate on this!

Author response: We agree that saline sand is not a good term and it has been changed throughout the manuscript.

The increase in T2* in the saltwater saturated sand cluster is surprising. Likely, this effect stems from the lithology in which the saltwater is intruding into is mostly different from the sand aquifer found in other soundings. This less compacted sand could be associated with larger pores and thereby an increase in T2* not directly linked to the saltwater.

Introduction/Methodology -> Context of research:

- Clearly there are other papers already dealing with joint interpretation of electrical methods and SNMR to solve the ambiguities in hydrogeophysical tasks (saltwater - clay, or vadose zone - clay). I think those should be mentioned. For instance to name just one - there are others as well:

Guenther and Müller-Petke, 2012, Borkum …

- Furthermore it is also necessary to point out joint inversion approaches that are in particular used to get joint layer boundaries for the interpretation. This is also necessary to point out that SNMR demands a resistivity distribution. There are different approaches around, for instance you may refer to:

Behroozmand

Skibbe

Author response: We have rephrased parts of the introduction and included more papers in this subject.

P4L9-L12

*"Others have used a joint-inversion approach where layer boundaries are set using multiple geophysical methods (Günther and Müller-Petke, 2012; Behroozmand et al., 2012). The joint approaches have the ability to delineate layer boundaries, not seen when inverted separately."*

- SNMR may also be able to detect water in partly saturated sands with fast relaxation. The is research of SNMR in the vadose zone and soils. For instance:

Flinchum, B. A., Holbrook, W. S., Parsekian, A. D., & Carr, B. J. (2019). Characterizing the critical zone using borehole and surface nuclear magnetic resonance. Vadose Zone Journal, 18(1), 1-18.

Walsh, D., & Grunewald, E. (2012, January). Application of surface NMR measurements to characterize vadose zone hydrology. In Symposium on the Application of Geophysics to Engineering and Environmental Problems 2012 (pp. 229-229). Society of Exploration Geophysicists.

Hiller, T., Costabel, S., Radić, T., Dlugosch, R., & Müller-Petke, M. (2021). Feasibility study on prepolarized surface nuclear magnetic resonance for soil moisture measurements. Vadose Zone Journal, 20(5), e20138.

Discussion:

Appears more a like a repetition of already mentioned statements.

From a methodlogical point of view I would expect discussion on pro/cons compared to existing approaches of joint inversion/interpretation especially having the comments above on clustering in mind.

Also discuss why not or how to combine a joint inversion with clustering, for instance using a layered joint inversion.

Author response: We have added a paragraph on joint inversions and how we opted for the separate inversions to use the spatial distribution of TEM to try and track boundaries more spatially.

P28L25-L35
*"Another way of exploiting collocated datasets is the use of joint inversion for layer boundary picking. Studies identifying layers from SNMR and TEM implementing various regularization techniques has shown promise in reducing the ambiguity found when interpreting each separately (Behroozmand et al., 2012); Skibbe et al., 2018). These approaches focus mostly on the collocated datasets and invert these jointly. In our study, the tTEM data is inverted separately with the full survey of more than 23000 datasets. As such, we have the ability to track the changes in resistivity in places where the SNMR is not present. Additionally, the framework for using joint inversion in steady-state SNMR is not established as kernels are calculated before the inversion, fixing the discretization. Further investigations could focus on implementing clustering in a joint inversion framework with a large spatial extent. This could alleviate some of the interpretational load when dealing with large datasets."*

It seems as only T2' is used here and might results in similar values even for different units. Why not using T2 or T1 as possible with the APSU device?

Author response: Yes, only T2* has been used in this manuscript for the clustering. At the moment, T1 is fixed and set equal to T2 in the inversion framework. We decided to use T2* as at the time of writing the resolution capability of T2 in steady state had not been demonstrated. Additionally, the T2 does at these sites not add a lot of different information for the clustering.

Some minor text comments:

page 2 line 23: replace inconsistent by ambigious
Author response: Fixed

page 3 line 1: cannot distinguish: this is a bit harsh. there are papers around who deal with partly saturated NMR stuff (see above)

Author response: Sentence has been rephrased to say that it is difficult to distinguish these scenarios.

P3L4-L7
"*As such, SNMR has difficulties distinguishing unconfined aquifers from semi-confined or confined aquifers without supplemental data, as the increase in water content cannot be established to be a saturation or a lithological transition (Behroozmand et al., 2015), Fig. 1.*"

page 3, second paragraph and figure 1: this is exactly what is described in (Günther, T., & Müller-Petke, M. (2012). Hydraulic properties at the North Sea island of Borkum derived from joint inversion of magnetic resonance and electrical resistivity soundings. Hydrology and earth system sciences, 16(9), 3279-3291.). Please cite or refer to it. It is a kind of perfect paper that lays ground for your paper as you go beyond this manual interpretation by your cluster analyses.

page 5 line 13: give numbers to deadtime

Author response: Fixed

page 5 line 14: define partly or fully decay

Author response: Added a description of this.
P6L1-L3

"*Signals from very small pores can therefore partially or fully decay, i.e. lose its amplitude and coherency, before the instrument has begun recording data.*"

page 5 paragraph 2.3: how is resistivity for MRS inversion handled. Any coupling there to the TEM?

Author response: The resistivity structure from the nearest TEM sounding is used for kernel calculations for the MRS inversion and is fixed. (Stated in P5L30)

page 11 table 1: as above, 1000 ohmm for saturated sand appears too high (cluster 1) and the overlap to cluser 3 in terms of resistivities is also high. this needs to be explained.

Author response: The very coarse material does have quite a high resistivity. But it is hard for the TEM to map the difference between a 300 ohmm layer and a 800 ohmm layer. Therefore, the high resistivities could be a product of the lack of sensitivity in the coarse unit.

We have added a description of this before the table.

P13L11-L13

"*The very high ρ (above 300 Ωm) is a product of very coarse material and that the TEM method can have limited sensitivity to determine resistivity above 150 Ωm.*"

page 16, table 2: t2* times of the saltwater sand and the freshwater sand are quite different. why?

Author response: As mentioned above, the newly deposited material in which the saltwater is intruding into, probably has a larger pore size than parts of the deeper aquifer which is mostly deposited by glaciers, likely has some remnant clay content.

page 16 line 17: saline sand -> replace by sand saturated with saline water (or something similar but saline sand is not correct)

Author response: This has been corrected in the manuscript.

---

## Author Comment (AC2)

Reviewer 2

The work that is presented in this manuscript is an important step towards an automated and thus fast hydrogeological interpretation of combined TEM and SNMR measurements. This combination has been applied in previous studies and has been proven to be powerful, especially within the frame of groundwater characterization in coastal areas, where the risk of relevant saltwater intrusion appears and frequently generates problems of freshwater production. In this environment, advanced knowledge of the aquifer situation, hydrogeologic modelling and temporal observation of the groundwater quality is desired and can most oftenly not be realized by drilling boreholes alone.

Recent improvements of TEM and SNMR have enabled options to produce big datasets that need to be interpreted in time to gain real social, economic and ecologic relevance in supporting freshwater production. Consequently, automated approaches such as the one presented in this study are the next logical step to deal with such data. The paper represents a significant contribution within the field of hydrogeophysics and its application and consequently fits in very well with the HESS journal.

The manuscript is well structured and the principle path to the results and conclusions is easy to follow. The figures are well presented and helpful to understand the key messages. However, some important technical information and discussions are missing, especially on details of the implementation and performance of the K-means approach. As I understand the study, the focus is not on the verification of the approach, e.g. by concrete ground truthing which is difficult given the large spatial extent covered by the surveys, but rather the introduction of the principal strategy that has still potential to be further developed. Please read my detailed concerns and recommendations below.

In the presented case studies, the outcome of the clustering analysis is basically tested against the classical manual interpretation at the basis of the original data with only a few observation boreholes. These are not sufficiently distributed over the profiles to enable reliable validation to the full extent. Consequently, the discussion section should honestly admit that there are still significant uncertainties, which cannot be resolved for several reasons. Please find details below.

Author response: We take this important point into consideration and we have added a section in discussion to highlight sparse borehole coverage.

However, the focus of this work and the value of automated clustering approaches in general is not on interpreting local situations as accurately as possible, even if the data at the specific position possibly allows this, but on providing a reliable overview at a larger scale. In this sense, I want to encourage the authors to include such reasoning in their motivation and discussion. Again, please find details below.

Finally, I assess the necessary revisions of the manuscript into the cathegory moderate-to-major.

Author response: Thank you for your comments. We have addressed the comments raised below and believe this has greatly improved the manuscript.

**Details**

**P2L12-13:**

... electrical properties of the subsurface, specifically the resistivity, are used ...

Author response: Fixed

**P2L22:**

A limitation of...

Author response: Fixed

**P2L23:**

The sentence starting with „An implication of ...“ is overloaded with redundancy and word repetitions. Please reformulate.

Author response: Sentence has been rephrased

P2L24-27

*"An implication of this is that local knowledge is required to link resistivity with the associated lithology or geological unit (Dickinson et al., 2010)."*

**P3L1:**

„...making it immeasurable with...“ => „...which makes the clay-bound water not detectable with...“.

Author response: Fixed

**P3L3:**

„established“ => „determined“

Author response: Fixed

**P3-Fig1:**

I like the idea of this scheme. However, I do not see the necessity of providing the axes as colorbars. It is rather confusing – I tried without any success to relate the font color inside the scheme to the colorbars of the axes. If you insist of using them, please clarify in the figure caption how they should be understood. Otherwise, I suggest avoiding them.

Please change the label of y-axis: „SNMR water content“ instead of „water content“ to avoid confusion or misconception. As you correctly describe in the main text, clay normally exhibits high water contents, but apparently appears as low or zero water content in SNMR.

Author response: We agree that the colorbars might be confusing. It was mostly to have presented the colorbars to the reader before they appear in the following figures. We have added a comment in the caption to indicate that the text colors are not related to the colorbar. We have added the new label.

Caption in Figure 1:

*"Figure 1 Different hydrogeological units resolved with TEM and SNMR. In dashed boxes, only one method is used, and the overlapping units show the ambiguities found. $T_2^*$ can be implemented to further separate units. Colors in text are not related to colorbars."*

**P5L5:**

Incorrect statement: the increased stacking rate leads to higher singal-to-noise ratios, not to higher signal amplitudes.

Author response: Fixed

**P5L11:**

Clarify with an additional brief sentence and a corresponding reference, why resistivity information is necessary for SNMR inversion.

Author response: Added a sentence and reference

P5L30-L32

"*Resistivity is needed to obtain the excitation fields used for kernel calculations (Braun and Yaramanci, 2008).*"

**P6L13:**

I am not familiar with K-means approaches. However, as far as I know, the performance of any optimization algorithm strongly depends on the chosen termination criteria. Please explain in more detail, here or later in the main text, how your approach actually performes. How many iterations were chosen exactly? Did you reach its maximum number for the majority of the runs or did the algorithm converge properly to the predeterimed minimum distance? Can you quantify the minimum distance? Are there any criteria how to choose the minimum distance or is it rather a try-and-error process to find a proper threshold?

Author response: The max iterations was never reached as the standard is 300 iterations. This is actually not a minimum distance but rather a tolerance. This means that if the centroid difference between two iterations is below a certain tolerance, the stopping criteria is met. This has been changed and better explained in the manuscript. We used the standard set of tolerance and maximum iterations from scikit-learn package.

P6L34-P7L2

"*The total distance from all data to their assigned clusters is then iteratively minimized through updating cluster center locations until either the centroid difference between iterations varies below a set tolerance or a maximum number of iterations is reached.*"

**P7L8 (subsection 2.5)**

This subsection, in my opinion, should appear at the very beginning of the methods section. Explain first what you did and where, before going into detail on each measurement method and processing approaches.

Author response: We believe that it the field description follows naturally after the methods as TEM and SNMR has already been explained at this stage. As the results section follows immediately after we have chosen to leave the order as is.

**P8L2**

Reading about „TEM soundings" here is rather confusing – it indicates that you conducted the „traditional" way of applying TEM. As I understand it, tTEM is rather a continuous sequence of TEM measurements while driving through the area. Please clarify.

Author response: Yes, indeed tTEM is a continuous sequence of TEM measurements. With soundings here, we mean every location linked to a single TEM model. For clarification we are writing TEM data now.

**Subsections 2.5.1 and 2.5.2:**

You already provide results, their interpretation and some conclusions here, while introducing the investigation areas. This is Okay in principle, if all this information has already been puplished. In this case, please specify the corresponding references. Otherwise, these statements belong to Section 3.

Author response: Only the Endelave NMR dataset has not previously been published. We have added a reference to the Endelave section for the TEM data.

P8L31-L33

*"The TEM data were acquired in April 2022 and cover the majority of the island and show ρ below 150 Ωm for the entire area (McLachlan et al., 2025)."*

**Subsection 3.1.1 (on the description of the silhouette index analysis and interpretation of Fig. 3)**

Unfortunately, some explanations in this passage remain unclear to me.

What exactly is the average silhouette index? Following the main text, it seems to be calculated for each cluster separately. But this is in contrast to the gray dashed lines in Fig.3 that represent a threshold for the whole data points in each subfigure. Moreover, it seems to be the very same value for each subfigure, no matter how many clusters are chosen - Is this plausible? Please clarify.

Author response: Thank you for this comment. From the other review, we have changed this figure to include the value of the average silhouette index in each subfigure. The average is calculated for each of the clustering runs (i.e., different for each subfigure). This value is used as a criterion for picking the number of clusters, the larger average silhouette index, the better.

Furthermore, when having a closer look at Fig.3, I cannot follow the classification of „well-, fairly and poorly defined" that is made in the main text. It states for instance that cluster 1 in Fig.3b was well-defined (P8L22-24) but more than half of the data points in that cluster are below the average value, some points even appear with a silhouette index below 0. Following the definition in the main text this should be classified as a poorly defined cluster (P8L20: „...many data points below the average..."). Similar confusion appears for the interpretation of the other subfigures. However, maybe I had missed something relevant and completely misunderstood this analysis. Please clarify and reformulate the text accordingly.

Author response: This was also raised by the other reviewer. The definitions of well, fairly and poorly-clustered has been added to the clustering section in methods. And yes, for Fig 3b, cluster 1 is only fairly defined with between 30-50% of datapoints above the average silhouette index.

P7L21-L26

*"The preferred number of clusters is chosen based on two criteria. Firstly, the highest average silhouette index as indicates that datapoints in general have the highest membership score with the given number of clusters. Secondly, we look at each cluster and their silhouette index. If*

*more than 50% of the cluster is above the average silhouette index, the cluster is well-defined, between 30-50% the cluster is fairly defined, and below 30% it is poorly defined.*"

**P10-Fig4**

Please specify the appreviations of the legend entries in the figure caption or provide a reference to table 1.

Author response: We have added a reference to table 1

**P11-Fig5 (and the other figures with profile data)**

Please consider indicating <N>orth and <S>outh directions at beginning and end of the profile(s) to make it easier to follow the interpretation(s) in the main text.

Author response: We have added a note in the figure caption to direct the reader.

**P12L11**

Following Archie's law, a change in water saturation should also lead to a contrast in rho. The fact that the change of water content measured by SNMR is not indicated by the TEM data is most likely due to the fact that TEM is not very sensitive to rho contrasts above 100 Ohm*m. This information should be given beforehand together with a suitable reference, e.g. when describing the scheme in Fig.1.

Author response: This was also raised by the other reviewer and a description of this has been added. It is likely due to the lack of sensitivity contrasts in TEM and not a lack of contrast in the resistivity itself.

P13L11-L13

"*The very high ρ (above 300 Ωm) is a product of very coarse material and that the TEM method can have limited sensitivity to determine resistivity above 150 Ωm (Christiansen et al., 2006).*"

**P12L13**

I do not doubt this statement, but you should provide a reference to verify it. There are some papers around that study the relationship between water saturation and relaxation time for loose sediments.

Author response: We have added a reference on this (Falzone and Keating, 2016).

**P13-Fig6 (same for Fig.10 and 11)**

I recommand focusing on T2* alone in the lower subfigure(s) to avoid confusion. My first reaction was wondering what additional information the ratio of T2*and T2 could provide. I guess there is no additional information in the T2 parameter compared to T2*, right?

Author response: Yes you are right. The T2 parameter does not hold extra information for this data set. Our focus is not T2 here, but for transparency we have elected to show both as we are inverting for it. Future implementation could have T2 as the main relaxation parameter for clustering.

**P13L14**

Please describe more in detail how the error bars in Fig.6 must be understood and how the uncertainty in assigning the data to the clusters affects the uncertainty of the water table estimation. What means „the uncertainty bars are based on the layer thickness"? Is it identical with the layer thickness? In this case, one could assume that, in general, we just have to decrease the layer thicknesses in the inversion model to increase the accuracy of the water table estimation. This would be nonsense, of course!

Author response: The uncertainty is assigned based on the layer thickness at the groundwater table depth. Since the layers are increasing in thickness, we expect the water table depth to be less accurate if not directly coinciding with the discretization. And yes, the error bars are directly the thickness of the layer in which the water table is found. As you mention, decreasing the layer thickness would not increase accuracy as the increased thickness reflects the decreased sensitivity at depth.

P17L5-L8:

*"The red line has a slope of 1 and the uncertainty bars are equal to the inversion layer thickness at the transition depth, as the clustering method is ternary (i.e., it has three options) and consequently, some layers found at cluster transitions could be assigned to either cluster."*

Sorry for stressing this aspect, but the confidence of non-invasive geophysical water table estimates is, in my experience, a frequent question in technical discussions with hydrogeologists. Your Figure 7 could be a key figure for the combined TEM/SNMR approach in this regard. However, it is not yet clear how to interpret it exactly.

And there is another aspect to consider when interpreting Fig.7. The water table is never identical with the interface between saturated and unsaturated zones. The saturated zone is always above the water table due to capillary forces. Depending on the lithology the difference between the two varies between a few cm (coarse sand) to several meters (till). Is such an interpretation of the descrepancy between the data points and the 1:1 line also reliable, given the actual geology?

Author response: As geology is mostly coarse sands this effect might be minimal in these cases. Its more plausible that the discrepancy originates from the smoothing of inversion parameters and the brute thresholding from the clustering.

**P13L21 and Fig.7**

I strongly recomment erasing the two yellow points from the crossplot. The attempt to verify the recent water table estimates with completely outdated information has no scientific value, even if it matched the 1:1 line by chance - which it obviously does not!

Author response: These data points were included because of availability but we have now deleted them as requested.

**P13L12**

Why do you not show the silhouette index plot for the Endelave site as well? Even if it was more complicated and difficult to interpret than for the case in Fig.3, the reader could learn a lot from it! How objective can the choice of the number of clusters even be? If an objective approach fails, we have to discuss criteria for subjective choices. As the number of clusters is crucial for this kind of analysis, I would expect a more detailed discussion on it for the two cases presented in this paper.

Author response: This decision was made to decrease the number of figures in the paper. We have added a section and the corresponding silhouette index figure for Endelave.

P19L3-L19

*"As before, we start by selecting the appropriate number of clusters, through silhouette index analyses, shown in **Error! Reference source not found.**. Considering we expect a more heterogeneous geology, three to six clusters are used in the analysis. In **Error! Reference source not found.**, three clusters are used to partition the data, and result in one well-defined, one fairly-defined and one poorly-defined cluster, whereas the yellow has low and even negative silhouette values, indicating wrongly assigned data points. The average silhouette index is the highest found with the assigned clusters. By using four clusters in **Error! Reference source not found.**b, two are well-defined, one fairly and one poorly clustered. We see less negative silhouette index data here, while still maintaining a high average silhouette index. Further increasing the number of clusters to five reveals similar silhouette indexes but has two fairly-defined clusters, however the average silhouette index drops, see **Error! Reference source not found.**c. Using six clusters is similar with a few well-defined and fairly-defined, and with a lower average silhouette index. The silhouette analyses show that the number of clusters should either be three or four as they have well-partitioned clusters, with the highest silhouette index. Prior information from the area indicates that we have four distinct geological units: tills, sand aquifers, Paleogene clay, and possible saline intrusion into sand. The blue cluster in **Error! Reference source not found.**b was found to have important hydrogeological information, regardless of its low silhouette index and, as such, we used four clusters for further results."*

**P13L13**

I recomment to erase the sentence „Consider Figure 8,...". The wording is technical incorrect, because the clustering results cannot be represented by two parameters. It is always three parameters regardless of how they are depicted. However, the sentence is not necessary at all.

Author response: The sentence has been deleted.

**P14L17**

The red cluster might also include unsaturated sand or can you exclude this possibility for some reason?

Author response: Since all resistivities in the cluster is 120ohmm or below, it would be a quite conductive unsaturated sand. The till is somewhat sand rich, so the distinction between unsaturated sand and the unsaturated sandy till is limited.

**P14L19**

saturated sand aquifers

Author response: Fixed

**P14L22**

„could indicate" => „indicates"

Author response: Fixed

**P15L2**

„would indicate" => „indicates"

Author response: Fixed

**P15-Fig8**

Please specify the appreviations in the legend.

Author response: Similarly to the previous comment, the table in which the abbreviations are stated has been referenced.

**P16L5**

Okay, we learn later that unsaturated sand at this position is somewhat unreliable because of the borehole information. However, I miss a discussion in the paper about the resolution properties of the tTEM at shallow depths < 10 m. This is relevant for the interpretation of the near-surface till layer. TEM does obviously not resolve it, although one would also expect low resistivities here.

Author response: Some of these boundaries are difficult to identify in resistivity only as we have reduced resolution in high resistivity contrast limits mentioned previously. In this case, it is potentially a lack of resistivity contrast between the sandy aquifer and the till underlying it, not necessarily the TEM being insensitive to the shallow depths.

**P17L8**

Figure 9a

Author response: Fixed

**P17L15**

What does „due to shifts in geological deposits" mean in this context? Please explain more in detail.

Author response: We agree that this phrasing is a bit tedious. There is some lateral changes in geology as indicated by the boreholes in this section, some showing sand, others till. Since the layer boundary from till to sandy aquifer matches the water table, this boundary is both the water table and a lithological boundary. We have deleted the last part of the sentence.

P23L15-17

*"This is interpreted as a semi-confined system with the water table coinciding with a lithological layer boundary."*

**P17L20**

A few boreholes? There is only one borehole in the vicinity of sounding 1 and 2. Furthermore, I would expect a decrease in rho at the SA-TI interface. Again, this contrast can obviously not be resolved by the tTEM method in this specific case. However, in Fig. 6 we see that it resolves a rho contrast between SA and TI even at a similar depth. Why not here?

Author response: Yes, we have changed it to state that it is a single borehole. There is a small change between these two resistivities, but the main driver of the different clustering is the NMR

parameters. It is worth noting that the resistivity contrast between these two in Figure 6 is substantially larger and is therefore more clearly visible in the figure.

**P18-Fig10**

The reddish colors of Silt and Clay/till are difficult to distinguish in the figure. Please use another color for one of them.

Author response: The silt color has been changed in both sections to be consistent with color scheme.

**P19L4**

Please be precise! The organic matter does surely not lead to a drop in WC (the opposite is the case!), which is very obvious when inspecting the cited paper of Mashhadi et al. (2024). Similar to clay - also with gyttja it is actually not a „real" but an apparent drop in WC caused by the fact that short T2*/T2 times are not detectable with SNMR.

Author response: Yes, it is the SNMR detectable WC which drops and not the actual WC. We have rephrased this.

P25L7-L9

"*The gyttja layer found in the borehole coincides with a drop in the SNMR WC due to the increases in organic matter, decreasing the pore size, and was grouped with the Cl-cluster (Mashhadi et al., 2024).*"

**P19L18**

„might have" => „might cause"

Author response: Fixed

**P20L8 – P21L6**

This passage is redundant with the descriptions in the introduction and can be removed from the discussion section completely. Please only discuss here what is new and implied by the direct results of your study.

Author response: We have rephrased and shortened this section to not be as long to shortly summaries the findings before proceeding to the specific discussion.

P27L11-16

"*In this study we investigated the use of clustering to combine the analysis of two geophysical methods, SNMR and TEM. The K-means clustering was found to be able to differentiate units into interpretable hydrogeological layers and was consistent with manual interpretations. Combining the datasets helped alleviate some of the ambiguities found when interpreting based only on a single dataset, i.e., unsaturated/confined conditions in Kompedal, and saltwater/freshwater in Endelave.*"

**P21L20**

As already mentioned (comment on **P12L13**), there are references to verify this statements. This is not a conclusion or finding of your study.

Author response: Reference has been added.

**P21L23**

„confidence in" => „confidence of"

Author response: Fixed

**P2L26**

„reduce" => „reducing"

Author response: Fixed

**P2L28**

„describe the most variance" => „describe most of the variance"

Author response: Fixed

**P21L32**

„informed" => „qualified"

Author response: Fixed

**P22L5**

provides

Author response: Fixed

**P22L29**

I strongly doubt that the suggested approach is strictly „non-subjective", because of the crucial predefinition of the number of clusters (please see my comments on the silhouette index approach). Use „less-subjective" or a formulation such as „towards a non-subjective interpretation".

Author response: We have rephrased.

P29L23-L26

"*K-means clustering of complementary SNMR and TEM models is shown to provide a less-subjective approach, where enhanced hydrogeological interpretations can be formed by exploiting the complementary nature of two data types.*"

**P23L3**

I disagree with this statement! Only for one of the site, the silhouette index approach was introduced and described, at all, and even this is hardly comprehensible (see my comments on section 3.1.1). For the second site, the choice of the number of clusters seems rather arbitrary to me. Finally, no concluding remarks can be made about the robustness of the suggested approach. This is also true for the whole K-means algorithm as it is used here and remains an

objective for future research. The robustness could, for instance, be analysed in a pure synthetic parameter study.

Author response: We have rephrased this sentence. After introducing the silhouette index for Endelave, we believe it is now more appropriate.

P29L31-L34

*"A silhouette index-based approach, combined with the a priori knowledge of the likely number of lithological units present, was used to select the number of clusters and found to be suitable for the these datasets."*

---

## Referee Report (RR1)

Review of

Alleviating interpretational ambiguity in Hydrogeology through clustering-based analysis of transient electromagnetic and surface nuclear magnetic resonance data

**P1L31:** "...to improve resolution of..." => "...to improve the identification of..."

**P5L27** (After the reference to Larsen et al., 2000): This might be a suitable position to inform the reader briefly about T2 and T2\*, both being a product of the inversion of the steady state SNMR data. T2 is first mentioned in the results section without any explanation where it comes from. Instead, a short introduction somewhere in the methods section is needed.

**P6L6** The information that T2\* is linked to pore size should appear before discussing the dead time issue. I recommend copy-and-pasting the whole sentence to P6L1 after "...termination is not possible". The link to the very last sentence in this paragraph ("This can be used...") would get lost in this case, but this one not necessary anyway and can be removed.

**P7L8** on => of

**P9L15** "results section" => "methods section"

P11L4 in => and

**P13L3** First mentioning of T2 – This parameter should be introduced beforehand, please see my comment on P5L27.

P13L13 "...less interaction of excited hydrogen spins with the grain surfaces"

**P15L1** The sentence starting with "The two data points..." is a relic from the first version of the manuscript and has to be removed.

**P15L8:** on => of

**P15L8:** "...are based..." => "...represent..."

**P15L9:** "...to provide a type of uncertainty." => "...to demonstrate the uncertainty related to the discretization of the resulting model."

**P15L14**: Figure 8a

**P20L16** Figure 10b

**P22L17** "when interpreting based only on a single dataset." => "when interpreting one of the individual datasets alone."

**P23L17** "...resulting in interactions leading to..." => "...resulting in a faster exchange of energy between excited hydrogen spins and pore walls leading to..."

P23L30 "which proved sufficient in ..." => "which has proven to be sufficient in ..."

**P24L12:** Remove the extra bracket

**P24L29:** "possible better to delineate" => "possible to better delineate"

---

## Author Response (AR3)

Editor comment: Please place the data in a repository (e.g., figshare or zenodo, etc.) alongside a README file containing an explanation, add the citation and/or doi and revise the data statement. You may add something like "assistance with reading the custom SNMR data is available upon request".

Author response: We have uploaded models to Zenodo and provided a new data statement.

**Reviewer 1**

General assessment after revision: All of my concerns about the first version of the manuscript are adequately answered and revised. Most of my previous recommendations on the presentation and discussion of the results are taken into account. If not, the authors provide adequate and comprehensible arguments for not doing so, which is fine. In particular, I appreciate the new passages in the discussion section and the extended explanations on the silhouette index including its application for the Endelave case study. The paper has much potential to motivate and inspire similar approaches on improving and partly automating the combined TEM/SNMR interpretation in future hydrogeology projects. My opinion is that the manuscript, after a few minor changes, is ready for final publication without another round of review.

The attached file contains a short list of typos that I noticed while reading and recommendations to (hopefully) improve clarity of a few specific formulations.

P1L31: "...to improve resolution of..." => "...to improve the identification of..."

Author response: Fixed

P5L27 (After the reference to Larsen et al., 2000): This might be a suitable position to inform the reader briefly about T2 and T2\*, both being a product of the inversion of the steady state SNMR data. T2 is first mentioned in the results section without any explanation where it comes from. Instead, a short introduction somewhere in the methods section is needed.

Author response: We have added a sentence to shortly describe T2 and that we are inverting for it alongside the water contents and T2\*

After Larsen et al., it now reads: "The data are also inverted for  $T_2$ , more directly linked to pore geometry, but is not used in the subsequent clustering."

P6L6 The information that T2\* is linked to pore size should appear before discussing the dead time issue. I recommend copy-and-pasting the whole sentence to P6L1 after "...termination is not possible". The link to the very last sentence in this paragraph ("This can be used...") would get lost in this case, but this one not necessary anyway and can be removed.

Author response: Fixed

P7L8 on => of

Author response: Fixed

P9L15 "results section" => "methods section"

Author response: Fixed

P11L4 in => and

Author response: Fixed

P13L3 First mentioning of T2 – This parameter should be introduced beforehand, please see my comment on P5L27.

Author response: After adding the description in the methods section this is not the first mention of T2.

P13L13 "...less interaction of excited hydrogen spins with the grain surfaces"

Author response: Fixed

P15L1 The sentence starting with "The two data points..." is a relic from the first version of the manuscript and has to be removed.

Author response: Fixed

P15L8: on  $\Rightarrow$  of

Author response: Fixed

P15L8: "...are based..." => "...represent..."

Author response: Fixed

P15L9: "...to provide a type of uncertainty." => "...to demonstrate the uncertainty related to the discretization of the resulting model."

Author response: Fixed

P15L14: Figure 8a

Author response: Fixed

P20L16 Figure 10b

Author response: Fixed

P22L17 "when interpreting based only on a single dataset." => "when interpreting one of the individual datasets alone."

Author response: Fixed

P23L17 "...resulting in interactions leading to..." => "...resulting in a faster exchange of energy between excited hydrogen spins and pore walls leading to..."

Author response: Fixed

P23L30 "which proved sufficient in ..." => "which has proven to be sufficient in ..."

Author response: Fixed

P24L12: Remove the extra bracket

Author response: Fixed

P24L29: "possible better to delineate" => "possible to better delineate"

Author response: Fixed

Reviewer 2:

Thanks for the revision.

The authors made a good job adressing my questions and issues. I recomand publication with some minor final corrections.

Final remark: Please check for consistency in phrasing: Sometimes you talk about silhuette index, other times it is scores or total average silhuette scores while in the caption it is just average value.

Author response: All phrasing of silhouette has been set to silhouette index for consistency.

**Reviewer 3:**

The authors appear to have taken care to implement the changes requested by the two reviewers who participated in the 1st round. This has, notably, improved the description of the methodology, which is the main point of novelty. I have just a couple reservations with respect to the terminology and definitions appearing, for example, in Figure 1.

Are we meant to believe that clay (presumably saturated, given that "unsaturated" is indicated for one of the porous media types in the figure) has a water content of <10%? This seems wrong. In my experience, clays often have porosities exceeding 50%. Did the authors mean the (saturated) "drainable" porosity or something of this sort?

Author response: Yes, it is true that water content of clay can exceed 50%. Here, we are describing the SNMR sensitive portion of the water within clays, which is extremely limited by the low pore sizes that water is residing in.

Also, echoing reviewer 1, "saline sand" is not a good term and has not been amended in the figure.

Author response: We agree with this and the wording has been changed to reflect the other changes to the manuscript.

In S3, I recommend swapping "fairly-defined" (strange English) for "moderately-defined" or "loosely-defined" or something like that.

Author response: We agree with this, and the wording has been changed to "moderately-defined".

As for the data availability statement, is "available upon request" coherent with EGU/Copernicus policies?

Author response: As the data from the SNMR is in a format made specifically for steady-state, it wont be readable by others. Therefore, the statement is kept as "available upon request" so that some help with reading this can be given.